# Effect of Dietary Pomelo Peel Powder on Growth Performance, Diarrhea, Immune Function, Antioxidant Function, Ileum Morphology, and Colonic Microflora of Weaned Piglets

**DOI:** 10.3390/ani12223216

**Published:** 2022-11-20

**Authors:** Yan Zeng, Xinrui Dai, Qingju Chen, Yubo Liu, Ziema Bumbie Gifty, Weizhong Sun, Zhiru Tang

**Affiliations:** 1Hunan Institute of Microbiology, Changsha 410009, China; 2Laboratory of Bio-Feed and Animal Nutrition, Southwest University, Chongqing 400715, China

**Keywords:** pomelo peel powder, antibiotic substitute, weaned piglets, growth performance, intestinal microorganisms

## Abstract

**Simple Summary:**

This research assessed dietary-accommodated pomelo peel powder (PPP) affected the growth performance, diarrhea, immunity, antioxidant capacity, ileum morphology, and colonic microflora of weaned piglets. Thirty piglets were weaned at day 28 and divided into three groups: basal diet (CON); CON accommodated with 75 mg/kg chlortetracycline; and CON accommodated with 8 g/kg PPP. These results showed that PPP increased growth performance, relieved diarrhea, and improved the intestinal microflora of weaned piglets. Therefore, PPP is expected to replace antibiotics as a feed additive to alleviate weaning stress and ensure the normal growth and development of piglets. This study provided a theoretical basis for PPP replacing antibiotics in weaned piglets’ diet.

**Abstract:**

This trial evaluated how dietary-accommodated pomelo peel powder (PPP) affected average daily feed intake (ADFI) and average daily gain (ADG), diarrhea, antioxidation, and colonic microbial in weaned piglets. Thirty piglets weaned at 28 d were divided into three groups: a basal diet (CON); a CON containing 75 mg/kg chlortetracycline (CTC); and a CON containing 8 g/kg (PPP). This trial had a period of 28 days. Piglets supplemented with PPP had higher ADFI and ADG than piglets in CTC and CON (*p* < 0.05). The diarrhea rate in PPP and CTC was lower than in CON in the 3rd and 4th weeks (*p* < 0.05). Serum superoxide dismutase and glutathione peroxidase enzyme activities, and total antioxidant capacity in PPP were higher than those in CON (*p* < 0.05). Serum interleukin (IL)-4, insulin-like growth factor-I, immunoglobulin (Ig)A, and IgG concentrations in the PPP and CTC groups were higher than those in the CON group (*p* < 0.05). Serum IL-1β, IL-8, IL-17, and interferon (IFN)-γ concentrations and the cecal pH in PPP were lower than those in CON (*p* < 0.05). Serum IL-1β, IFN-γ, and IgA concentrations of piglets in PPP were lower than in CTC (*p* < 0.05). The villus height and villus height/crypt depth of the ileum of piglets in PPP and CTC were higher than those in CON (*p* < 0.05), but there was no difference between PPP and CTC (*p >* 0.05). The Firmicutes and Cyanobacteria relative abundances in PPP and CTC (*p* < 0.05) were lower than those in CON, whereas the Bacteroidetes relative abundances in PPP and CTC were higher than those in CON. The Prevotellaceae relative abundance in CTC was higher than in CON (*p* < 0.05), whereas the Lactobacillaceae relative abundance in CTC was lower than in CON (*p* < 0.05). The Ruminococcaceae relative abundance in PPP was higher than in CON (*p* < 0.05), whereas the Veillonellaceae relative abundance in PPP was lower than in CON (*p* < 0.05). PPP can improve ADFI and ADG, relieve diarrhea, and enhance the colonic microflora of weaned piglets. Therefore, PPP is expected to replace CTC as a feed additive to alleviate weaning stress and ensure normal growth and development of piglets.

## 1. Introduction

Early and ultra-early weaning techniques are often used in modern pig production to improve the utilization rate of sow. The incomplete development of the digestive system and immune organs in weaned piglets, environmental changes, and dietary intake changes during weaning decreases nutrient digestion and absorption capacity and disease resistance, increases diarrhea rate, induces the “weaned stress syndrome,” and increases the mortality rate of piglets [1]. Although the use of antibiotics can effectively alleviate these problems, the use of a large number of antibiotics causes animal intestinal flora disorder, stronger bacterial resistance, and drug residues. Owing to the emergence of the breeding antibiotic-free era of livestock and poultry, it is urged to find safe and effective substitutes for antibiotics in weaned piglet production.

Pomelo fruit is an abundant and popular fruit. Pomelo (the family *Rutaceae*) is named pummelo or shaddock [2]. Pomelo is an important commercial cultivar of the Citrus genus. The FAO database in 2018 showed that the global yield of pomelo (including grapefruit) was 9.4 × 10^9^ kg, such as 5.07 × 10^9^ kg for China, 6.58 × 10^8^ kg for Vietnam, 5.59 × 10^8^ kg for USA, 4.60 × 10^8^ kg for Mexico, and 4.45 × 10^8^ kg for South Africa [3]. Pomelo fruit can be often processed into cans and juice. Pomelo peel accounts for 30~50% of pomelo fruit (*w*/*w*), so this consumption produces a huge amount of pomelo peel as a by-product. There are 2.0 × 10^6^ tons/year pomelo peel in China and 2.8~4.7 million tons pomelo peels for world production in 2018.

Some studies have found that pomelo peel powder (PPP) contains physiologically active ingredients such as flavonoids, limonin, and volatile oils [4,5,6]. These bioactive molecules play unique roles in animal growth and development, anti-inflammation, antioxidation, hypolipidemic, antibacterial, anticancer, and intestinal development [7,8,9,10,11]. Pomelo peel also contains many nutrients, such as natural pigments, minerals, vitamins, polysaccharides, pectin, and fiber. The fiber of pomelo peel powder is not digested in the front intestine but reduces blood sugar and blood lipids and prevents hypertension, hyperlipidemia, hyperglycemia, and colon cancer by large intestinal fermentation. An in-depth study of dietary fiber found that dietary fiber can improve the intestinal microbial community and its metabolites, maintain homeostasis of the intestinal environment, and promote the absorption of nutrients, thus improving intestinal health [12,13]. PPP can be added to feed to promote animal growth and health, thus converting waste into treasure. Pomelo peel is hopeful to achieve an economic sustainable development of animal husbandry in an antibiotic-free era.

However, few researches on the effects of PPP on antioxidation in livestock been reported. To provide a reference data for PPP replacing antibiotics in weaned piglets’ diet, PPP was accommodated to corn- and soybean-based diets of weaned piglets to study that PPP affected average daily feed intake (ADFI), average daily gain (ADG), antioxidant capacity, and colonic microflora of weaned piglets in this study.

## 2. Materials and Methods

### 2.1. Materials

PPP was provided by the College of Horticulture and Landscape Architecture (Southwest University, Chongqing, China). Pomelo peels were washed, cut into small pieces, and dried in oven (80℃), and then crushed for PPP. PPP was packed in a clean plastic bag for the piglet experiment. The PPP contained 86.36% dry matter, 8.23% crude protein, 2.81% ether extracted matter, 3.56%crude ash, 0.36% Ca, 0.93% P, 17.72% crude fiber. The PPP contained 1.56 mg/g flavonoids.

The biochemical kits for glutathione peroxidase (GSH-Px), superoxide dismutase (SOD), and catalase (CAT) enzyme activities, the biochemical kits of total antioxidant capacity (T-AOC) kits, the ELISA kits for interferon (IFN)-γ, interleukin (IL) -1β, IL-2, IL-4, IL-6, IL-8, IL-10, IL-17, insulin-like growth factor (IGF) -1, prostaglandin E2 (PGE2), immunoglobulin (Ig) A, IgM, and IgG were purchased from the Institute of Nanjing Jiancheng Bioengineering (Nanjing, China). The MOBIO power fecal DNA extraction kit was bought from MoBIO Laboratories, Inc. (Carlsbad, CA, USA). Illumina TruSeq DNA PCR-Free Prep and Illumina MiSeq Reagent kits were purchased from Invitrogen Corporation (Carlsbad, CA, USA).

### 2.2. Experimental Animals, Design, and Diets

Thirty piglets (Duroc × Landrace × Yorkshire, 7.48 kg ± 1.19) weaned at day 28 were obtained from a commercial pig farm (Chongqing, China). Piglets were randomly divided into three groups: a basal diet (CON); a CON supplemented with 75 mg chlortetracycline per kg diet (CTC); and a CON supplemented with 8 g PPP per kg diet (PPP). Each group had 10 barrows with one pig per replicate. The basal diet was designed according to NRC requirements [14]. Dietary compositions and nutritional level of three diets were presented in Table 1. This dose of PPP was based on 1g/kg body weigh/day for mice reported by the previous literature [15,16] with slight modifications. This trial had a period of 28 days.

Piglets were fed individually in room kept at a constant temperature (30 ± 1.2 °C) with pens (0.8 m depth × 0.5 m width × 1.5 m length), with intake water and diet ad libitum. Before feeding in the morning, piglets were weighed on day 0 and 28. Piglets’ feed intakes were recorded daily during the 28 days. Based on the scoring criteria of stool shape reported by Montagne et al. [17], piglets’ stool shape was recorded daily. Diarrhea incidence and diarrhea index were recorded daily in the whole trial period. This experimental process obeyed the requirement of Animal Experimentation Ethics Committee of Southwest University (IACAU-20171210-01).

### 2.3. Sample Collection

Five piglets were picked up from each group before piglets fed on the morning of day 28 and a 10 mL blood was sampled from the orbital sinus, left undisturbed for 60 min. Serum sample were harvested under a 3500× *g* centrifugation for 10 min at 4 °C and stored at −20 °C for biochemical analysis and ELISA. After blood collection, four piglets per group were intravenously injected 50 mg/kg body weight sodium pentobarbital to be anesthetized and exsanguinated. The 2 cm middle ileum tissues were washed with sterile saline and soaked in a 4% formaldehyde solution and 10 mL colonic contents were collect for microflora sequencing.

### 2.4. Growth Performance and Organ Indexes

(ADFI, ADG), F/G, and diarrhea rate were calculated using the following formulas:Total food intake/days =ADFI (g/d)(1)
(Final weight − initial weight)/days =ADG (g/d)(2)
Food intake/weight gain = F/G (g/g) (3)
Number of diarrhea pigs/(number of total pigs ×days) × 100 = diarrhea rate (%)(4)

### 2.5. Serum Antioxidant Indexes and Immune Indexes

According to kits’ instructions, serum enzyme activities of SOD, GSH-Px, and CAT, serum T-AOC were analyzed using a CL-8000 automatic autoanalyzer Shimadzu Company (Shanghai, China) and serum IL-1β, IL-6, IL-4, IL-8, IL-10, IL-17, IFN-γ, PGE2, Ig A, Ig M, Ig G, and IGF-1 concentrations were measured using porcine-specific ELISA kits.

### 2.6. Hematoxylin-Eosin Staining

The sliced ileum tissue samples were obtained hematoxylin-eosin staining and viewed under an optical microscope of Carl Zeiss Inc. (Oberkochen, Germany). Digital images of the sliced ileum tissue samples were captured by a Sony 3CCD-VX3 camcorder. The goblet cell, lymphocyte number, villus height, and crypt depth were measured by the image analysis software of Intronic GmbH & Co. (Rothenstein, Berlin, Germany).

### 2.7. The 16S rDNA MiSeqensing

The colonic microbial DNA was extracted by the instructions of MOBIO power fecal DNA extraction kit and colonic microbial 16S rDNA fragments were analyzed by Chengdu Luoning Biological Technology Co. (Chengdu, China). The V4 region of colonic microbial 16S rDNA was amplified by specific primers 515F (5′-GTGCC AGCMG CCGCG GTAA-3′) and 806R (5′-GGACT ACHVG GGTWT CTAAT-3′) with barcodes. The recovered PCR products were detected and quantified using the Qubit 2.0. A 16S rDNA library was constructed by a kit of Illumina TruSeq DNA PCR-Free Prep. Misequencing was performed using a kit of Illumina MiSeq Reagent. PE reads were obtained and spliced using FLASH software. After removing low-quality bases and contaminated joint sequences, data filtering was completed and analyzed using the UPARSE software. The function of colon microbial genes in piglets was predicted and analyzed using PICRUSt software 2.0 from Chengdu Luoning Biological Technology Co. (Chengdu, China).

### 2.8. Data Treatment and Statistical Analyses

Based on a completely factorial design, trial data were treated by the one-way analysis of variance of SAS 8.2 of SAS Institute, Inc. (Cary, NC, USA). The differences among the three groups were analyzed by the LSD test and *p* < 0.05 was shown significance. The mean ± standard error of the mean was to prescribe all data.

## 3. Results

### 3.1. ADG, ADFI, Diarrhea Rate, and Organ Indexes

As are shown in Table 2. Compared with CON and CTC, dietary PPP had higher final weight and ADG and ADFI (*p* < 0.05) and had no effect on F/G (*p >* 0.05). There was no difference between CON and CTC in ADG, ADFI, and F/G (*p >* 0.05). There were no differences among the three groups in liver, thymus, pancreas, and kidney indices (*p* > 0.05). The spleen index in PPP was lower than that in CTC (*p* < 0.05). There was no difference between PPP and CON in the spleen index (*p* > 0.05). There was no difference among three groups in the diarrhea rate on the 1st and 2nd weeks (*p* > 0.05). The diarrhea rate in PPP and CTC was lower than in CON by the 3rd and 4th weeks (*p* < 0.05).

### 3.2. Serum Antioxidant Indexes, Cytokines, and Immune Indexes

Serum antioxidant indexes, cytokines, and immune indexes of piglets were shown in Table 3, serum CAT and GSH-Px activities of piglets in PPP were higher than those in CON and CTC (*p* < 0.05), whereas there were no differences between CON and CTC (*p* > 0.05). Serum SOD and T-AOC enzyme activities of piglets in PPP were higher than those in CON (*p* < 0.05), whereas there were no differences between CON and CTC, or between PPP and CTC (*p* > 0.05). Compared with CON, dietary PPP increased serum IL-4, IgA, IgG, and IGF-I concentrations and decreased serum IL-1β, IL-8, IL-17, and IFN-γ concentrations (*p* < 0.05). Serum IL-1β, IFN-γ, and IgA concentrations in PPP were lower than those in CTC (*p* < 0.05). Serum IL-4, IgA, IgG, and IGF-I concentrations in CTC were higher than those in CON. The serum IL-8 concentration in CTC was lower than that of piglets in CON (*p* < 0.05). There were no differences among the three groups in serum IL-6, IL-10, PGE2, and IgM concentrations (*p >* 0.05).

### 3.3. Ileum Morphology and pH in Gastrointestinal Tract of Piglets

Ileum morphology and pH in gastrointestinal tract of piglets were shown in Table 4, piglets in PPP and CTC had lower pH of duodenal contents (*p* < 0.05), and piglets in PPP had lower cecal content pH than those in CON (*p* < 0.05). There was no difference among the three groups in the pH of the stomach, jejunum, ileum, and colon (*p* > 0.05). There was no difference among the three groups in ileal crypt depth (*p* > 0.05). The villus height and villus height/crypt depth in the ileum in PPP and CTC were higher than those in CON (*p* < 0.05), but there was no difference between PPP and CTC (*p* > 0.05). However, the intestinal villus height/crypt depth in PPP was higher than that in CTC (*p* < 0.05).

As shown in Figure 1, compared with CON, the ileal villi in CTC and PPP were clearly defined, arranged neatly and densely, and had fewer broken fragments, while the ileal villi in CON were seriously broken and lost, and the villous epithelial cells were exfoliated and necrotic.

### 3.4. Colonic Microflora of Weaned Piglets

#### 3.4.1. Sequencing Information of Colonic Microflora and Venn Diagram

In total, 781,764 pieces of colonic microflora sequence data with 316 bp average reading length were obtained. The microflora mainly consisted of Firmicutes (67.68%), Bacteroidetes (24.67%), Proteobacteria (2.98%), Tenericutes (1.55%), Cyanobacteria (1.21%), Spirochaetes (0.89%), and Actinobacteria (0.236%). As shown in Figure 2, The microbial communities were explored by a Venn diagram, indicating that the colonic microbial communities in three groups had 1185 OTUs in common, accounting for 44.1% (CON), 32.8% (PPP), and 33.4% (CTC) of the OTUs, and there were 572 species of characteristic flora in CON, 1303 species in CTC, and 1214 species in PPP.

#### 3.4.2. Bata Diversity Analysis and Alpha Diversity Analysis

As shown in Table 5, although there no difference in alpha diversity indices among three groups (*p* > 0.05), the Chao1, ACE, and Shannon in PPP were higher than those in CON (*p* > 0.05) (Figure 2).

#### 3.4.3. Composition and Structure of Microflora

As shown in Table 6, these phylum of colonic microbiota in three groups were mainly Firmicutes, Bacteroidetes, Spirochaetes, Proteobacteria, Tenericutes, Cyanobacteria, and Actinobacteria. Compared with CON, the Firmicutes and Cyanobacteria relative abundances in PPP and CTC decreased (*p* < 0.05), whereas the of Bacteroidetes relative abundances in PPP and CTC increased (*p* < 0.05).

As shown in Table 7, the families of colonic microbiota had Prevotellaceae, Veillonellaceae, Lachnospiraceae, Ruminococcaceae, Bacteroidales S24-7, Clostridiales, Lactobacillaceae, Paraprevotellaceae, and Spirochaetaceae. Prevotellaceae relative abundance in CTC was higher than that in CON and PPP (*p* < 0.05), while Lactobacillaceae relative abundance in CTC was lower than that in CON and PPP (*p* < 0.05). Ruminococcaceae relative abundance in PPP was higher than that in CON and CTC (*p* < 0.05), while Veillonellaceae relative abundance in the PPP group was lower than that in CON and CTC (*p* < 0.05).

#### 3.4.4. Function of Colon Microbial Genes in Piglet

As shown in Table 8, six metabolic pathways of colon microbiota were analyzed. There were no differences in genetic information processing, environmental information processing, human diseases, and organismal systems among three groups (*p >* 0.05). Compared with CON, piglets fed PPP and CTC showed increased cellular processes (*p* < 0.05). There was no difference between CTC and PPP in cellular processes. Compared with CON and PPP, dietary CTC increased metabolism (*p* < 0.05). There was no difference between CON and PPP in the metabolic function (*p >* 0.05).

Differences in the colonic microbial gene metabolism relative abundance of piglets are presented in Table 9. The carbohydrate metabolic function in PPP is higher than that in CON and CTC (*p* < 0.05), whereas were no significant differences between CTC and CON. Compared with CON and PPP, dietary CTC increased the degradation and metabolic function of exogenous substances (*p* < 0.05), whereas there was no difference between CON and PPP.

## 4. Discussion

### 4.1. Dietary PPP Affected Growth Performance and Diarrhea Rate

Growth retardation and diarrhea of weaned piglets are important problems in pig production. CTC is often used in pig production to alleviate piglet diarrhea [18]. Studies have shown that plant extracts such as flavonoids [4], pectin [5], and vitamins can relieve diarrhea in animals; pomelo peel contains bioactive substances such as flavonoids, pectin, and vitamins. The results in this study showed that the diarrhea rate in weaned piglets fed a CON was not relieved at the 3rd and 4th weeks. However, weaned piglets administered PPP and CTC significantly decreased the diarrhea rate, whereas there no significant difference in diarrhea between PPP and CTC, which indicated that PPP had the same effect as CTC in fighting piglet diarrhea. The essential oils in pomelo peel contain terpenes, aldehydes, and alcohol constituents (α-terpineol, α-pinene, α-phellandrene, β-myrcene, β-pinene, β-phellandrene, β-caryophyllene, ρ-cymene, citral, citronellal, linalool, and ocimene) which have antimicrobial activity [19]. The mechanism of these constituents in pomelo peel to inhibit bacterial growth owes to attacking the phospholipid bilayer of the cellular membrane, disrupting enzyme systems, damaging genetic substances of pathogenic bacteria [20].

The ADG and ADFI of weaned piglets fed PPP were significantly higher than those in CON, indicating that dietary PPP can effectively improve the loss of appetite in weaned piglets under weaning stress and relieve the symptoms of slow development caused by lack of food. Supplementation of PPP containing 1.56 mg/g flavonoids increased average daily gain of piglets, and decreased diarrhea rate. This result was consistent with previous studies reporting the addition of citrus flavonoids reduced the occurrence of zoonotic diseases and has a significant effect on improving animal performance [21,22].

### 4.2. Dietary PPP Affected Serum Antioxidant Enzyme Activities

Both GSH-Px and SOD can catalyze the reduction in cytotoxic peroxides or superoxide free radicals and protect cells, which is an important index for the evaluation of oxidative stress in the body. CAT catalyzes the decomposition of H_2_O_2_ and is a marker enzyme of peroxisomes, accounting for approximately 40% of the total peroxisome [23]. Various antioxidants and antioxidant enzymes in the body constitute T-AOC, and the T-AOC level reflects antioxidant capacity in the measured substances.

In this study, serum CAT and GSH-Px activities of piglets in PPP were significantly higher than those in CON and CTC, whereas there were no significant differences between CON and CTC. Serum SOD activities and T-AOC of piglets in PPP were significantly higher than those in CON, whereas there were no significant differences between CON and CTC, or between PPP and CTC. This result suggests that oxidative damage caused by weaning of piglets can lead to a decrease in T-AOC and the activities of GSH-Px, CAT, and SOD, and the antioxidant capacity of PPP is higher than that of CTC. Here, PPP shows stronger antioxidant capacity because more antioxidants are released in PPP under digestion of piglets. The bioactive substances in PPP, such as naringin, naringin, and volatile oils, can enhance the antioxidant capacity of the body by increasing the activity of antioxidant enzymes [7,16]. CTC alleviates stress by inhibiting bacterial invasion, and SOD catalyzes the disproportionation of superoxide anion radicals to produce O_2_ and H_2_O_2_ [24].

### 4.3. Dietary PPP Affected Serum Cytokines and Immunoglobulin

IL-1β can promote T-cell activation, costimulatory antigen-presenting cells, B-cell proliferation, and antibody secretion [25]. IL-8 attracts and activates neutrophils. After binding to IL-8, neutrophils activate displacement and release inflammatory mediators, leading to a local inflammatory response [26]. IL-17 can induce endothelial cells, epithelial cells, and fibroblasts to secrete PGE2 IL-6 and IL-8 [27]. IFN-γ has antitumor, immunomodulatory, and antiviral properties and activate APC and promote Th1 cell differentiation by upregulating transcription factors [28]. In this experiment, we observed that dietary PPP significantly reduced pro-inflammatory cytokines (IL-1β, IL-8, IL-17, and IFN-γ) caused by weaning of piglets. Hu et al. [16] observed that dietary citrus Peel Powder significantly reduced the increase in pro-inflammatory cytokines (MCP-1, IL-6, and TNF-α) caused by HFD. Liu et al. [29] also suggested that citrus peels can reduce pro-inflammatory cytokines (IL-6, TNF-α, and MCP-1) to exhibit anti-inflammatory activity, which is consistent with our results. The anti-inflammatory cytokine IL-4 in PPP was significantly higher than in CON, indicating that dietary PPP alleviated the occurrence of inflammatory reactions in the body. IL-4 can promote Th2 cells and exert an anti-inflammatory effect [30]. The bioactive substances in PPP can participate in anti-inflammatory reactions in the body [31].

Immunoglobulins are proteins secreted by immune cells. They have immunomodulatory functions and can bind antigens to resist pathogenic infections. In this experiment, PPP can significantly increase serum IgA and IgG content of piglets, indicating that PPP can produce more immunoglobulin in body and enhance immune response, indicating that PPP can improve the immunity of the body. The active components such as flavonoids and volatile oil in pomelo peel have a good effect on improving the immunity of piglets [32].

IGF-1 lowers blood sugar and blood lipids, relaxes blood vessels, and promotes cell growth and differentiation [33]. It was found that naringin had a hypolipidemic effect in hyperlipidemic mice; however, in this experiment, the IGF-1 level in PPP was significantly higher than that of CON, indicating that PPP had an effective active substance that promotes the synthesis of IGF-1 in weaned piglets, which may indicate that pomelo peel can also have the same hypolipidemic effect in pigs.

A lot of bioactive compounds in CPP have an important role in the health of the organism. Citrus fruit peel contains essential oils, pectin, and polyphenol [29]. Limonene and the main component in citrus essential oil have antibacterial and antioxidant activity [34]. Pectin in citrus fruits is effective in improving intestinal inflammation [29]. The active components, such as flavonoids and volatile oil, in pomelo peel have a good effect on improving the immunity of piglets and reducing the inflammatory reaction, thus reducing the diarrhea rate of piglets.

### 4.4. Dietary PPP Affected Intestinal Morphology and Intestinal pH of Weaned Piglets

This study showed that ileum villus height and V/C ratio in CTC and PPP were significantly higher than those in CON, which may indicate that PPP and CTC can increase the development of the ileum tissue of weaned piglets. This may be due to the fact that PPP and CTC can maintain an intestinal epithelial cell barrier and alleviate intestinal inflammation caused by weaning of piglets and PPP contains pectin which protects the integrity of intestinal mucosa. The height of the ileum villi reflects the absorption ability of the small intestine, and the decrease in villus height indicates that the absorption ability is low, and the depth of the crypt reflects the quality of the secretory function of the small intestine. A decrease in the V/C ratio indicates the low digestion and absorption ability of the ileum with diarrhea [35].

The results showed that PPP and CTC can significantly reduce the pH value of the duodenum and cecum content in piglets, indicating that PPP and antibiotics can reduce the normal function of intestinal pH, maintain the stability of intestinal microorganisms in piglets, promote the absorption and utilization of nutrients in piglets, and promote the growth and development of piglets. pH is one of the most important factors for maintaining homeostasis in the internal environment of animals. Before weaning, breast milk was obtained from all piglets. After weaning, piglets were transferred directly from liquid breast milk to solid feed, feed intake decreased, lactic acid produced by lactose fermentation was limited, and insufficient gastric acid secretion led to an increase in gastrointestinal pH. Lactic acid produced by lactose fermentation is an important substance for maintaining low intestinal acidity [36]. The secretion and activity of pepsin, trypsin, chymotrypsin, and amylase are affected by intestinal pH [37]. The decrease in pH in the duodenum and cecum of weaned piglets in the CTC and PPP was attributable to the inhibition of pathogenic bacteria, and the promotion of beneficial bacteria, such as lactic acid bacteria of pomelo peel essential oils and CTC [34,38].

### 4.5. Dietary PPP Affected Colonic Microflora

The microbial barrier in intestine is formed by the settlement of intestinal microorganisms in the mucosa to prevent excessive reproduction of pathogenic bacteria in the intestinal segment [39]. Their main function is to maintain the balance of intestinal microecology, assist the immune cells and immune factors of intestinal mucosa to perform their functions, promote the development of intestinal immune system, and improve the immune ability of the body [40].

The nutrition of the intestinal flora comes from the dietary composition and exfoliated epithelial cells of the host, which is an organ with extensive metabolic capacity and substantial functional plasticity. In this experiment, the dominant phylum flora in the three groups contained *Firmicutes*, *Bacteroidetes,* and *Proteobacteria*. In this experiment, the *Bacteroidetes* relative abundance in PPP and CTC was significantly higher than those of CON, indicating that PPP and CTC can improve the colonic microbial of piglets. These characteristics of the intestinal flora make the focus of the study shift from the richness and diversity of microbial members to function. A clinical study by Olaisen et al. found that the *Bacteroidetes* relative abundance in patients’ intestines with Crohn’s disease and ulcerative colitis decreased, indicating that the abundance of *Bacteroidetes* was related to intestinal inflammation and immunity to a certain extent [41].

In this experiment, the *Firmicutes* relative abundance in PPP decreased significantly, indicating that PPP can control lipid synthesis and reduce fat synthesis and that PPP can be used to improve the lean meat percentage of pigs. The ratio of the relative abundance of *Firmicutes* to that of *Bacteroidetes* is related to obesity, and individuals with a high ratio are more likely to be obese [42]. Weaning oxidative stress can increase the intestinal *Cyanobacteria* relative abundance in weaned piglets [43]. However, in this experiment, the *Cyanobacteria* relative abundance decreased in CTC and PPP, indicating that the piglets in these two groups were relieved of their degree of weaning oxidative stress.

In this experiment, the dominant family bacteria in CON were *Veillonellaceae*, *Lachnospiraceae*, *Ruminococcaceae*, and *BacteroidalesS24-7*, while those in the antibiotic group were *Prevotellaceae*, *Veillonellaceae*, and *Lachnospiraceae*. The dominant bacteria in PPP were *Prevotellaceae*, *Lachnospiraceae*, and *Ruminococcaceae* species. The *Prevotellaceae* relative abundance in the antibiotic group was significantly higher than those in CON and PPP.

*Prevotellaceae* plays a major role in the digestion of starch, cellulose, and proteins [44], indicating that CTC can improve intestinal microorganisms and improve the digestion and absorption of nutrients in the intestinal tract. The *Veillonellaceae* relative abundance in PPP decreased significantly, and the *Veillonellaceae* relative abundance increased when the body was infected, while the *Veillonellaceae* relative abundance in PPP was significantly lower than in CON, the results showed that the intestinal infection of piglets in PPP was alleviated. *Ruminococcaceae* ferments cellulose and pentosans into organic acids, which can inhibit pathogenic bacteria growth in intestines and protect the intestinal tract [45]. The improvement in the *Ruminococcaceae* relative abundance in PPP indicated that dietary PPP to the diet can help maintain the stability of intestinal microorganisms and exert normal digestion and absorption function in the intestinal tract.

Through functional prediction of colonic microflora, it was found that the microbial community of piglets supplemented with PPP had relative advantages in cell metabolic function (such as cell activity, cell growth, and apoptosis) and body metabolic function (such as lipids, amino acids, and terpenoid metabolism). CTC has a relative advantage in terms of its cell metabolic activity. It has been suggested that PPP may improve the microbial community in cell life processes and body metabolism. The above results show that CTC and PPP can improve colonic microflora and have a positive impact on health of piglets through colonic microorganisms and their products.

## 5. Conclusions

Based on the effects of PPP or CTC on growth performance, diarrhea rate, antioxidant capacity, anti-inflammatory ability, immunity, ileum mucosal morphology, and colonic microflora stability in weaned piglets, the results suggest that PPP can replace CTC. Dietary administration of PPP improved growth performance and reduced the diarrhea rate, which attributed to antioxidant capacity improvement, anti-inflammatory ability, immunity enhancement, and colon microflora stability of PPP containing physiologically active ingredients, such as flavonoids, limonin, volatile oils, polysaccharides, dietary fiber, vitamins, minerals, pectin, and natural pigments. Therefore, PPP shows potential as a feed additive for early weaned piglets. Further studies are expected to elucidate the mechanism of various active ingredients of PPP affecting intestinal health as a feed additive.

## Figures and Tables

**Figure 1 animals-12-03216-f001:**
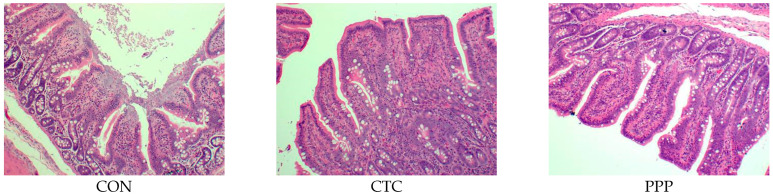
Dietary pomelo peel powder affected ileum morphology in weaned piglets.

**Figure 2 animals-12-03216-f002:**
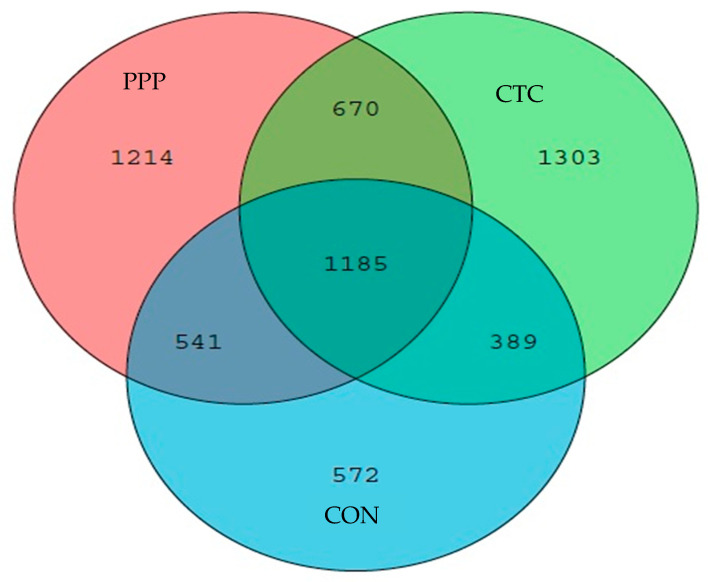
Venn diagram.

**Table 1 animals-12-03216-t001:** Dietary compositions and nutrition level (**%**, air-dry basis).

Ingredients	Treatments ^2^			Nutrition Level	Treatments		
	CON	CTC	PPP		CON	CTC	PPP
Corn	60.65	60.65	59.85	DE (MJ/kg)	14.30	14.30	14.3
Soybean meal	13.00	13.00	13.00	CP (%)	18.40	18.40	18.4
Puffed soybean	9.80	9.80	9.80	CF (%)	2.80	2.80	4.10
Fish meal	5.00	5.00	5.00	Available phosphorus (%)	0.40	0.40	0.40
Whey	6.00	6.00	6.00	Lys (%)	1.42	1.42	1.42
Fat powder	1.50	1.50	1.50	Met (%)	0.49	0.49	0.49
CaHPO_4_	1.00	1.00	1.00	Thr (%)	0.90	0.90	0.90
Limestone	0.63	0.63	0.63	Ca (%)	0.78	0.78	0.78
Choline chloride	0.20	0.20	0.20	Ca (%)	0.78	0.78	0.78
NaCl	0.30	0.30	0.30				
L-Lysine·HCl	0.54	0.54	0.54				
Met	0.19	0.19	0.19				
Thr	0.19	0.19	0.19				
Chlorotetracycline	0.00	0.0075	0.00				
Pomelo peel powder	0.00	0.00	0.80				
Premix ^1^	1.00	1.00	1.00				
Total	1.00	1.00	1.00				

^1^ Provided per kilogram of diet: 10,500 IU V_A_, 3000 IU V_D3_, 22.5 IU V_E_, 3.0 mg V_K3_, 4.5 mg V_B6_, 0.03 mg V_B12_, 15 mg pantothenic, 1.5 mg folic acid, 7.5 mg riboflavin, 30.0 mg niacinamide, 3.0 mg thiamine, 0.12 mg biotin, 100 mg Zn, 100 mg Fe, 4.0 mg Mn, 6.0 mg Cu, 0.3 mg I, 0.3 mg Se. ^2^ CON, the basal diet; CTC: the basal diet supplemented with 75 mg/kg chlortetracycline; PPP: the basal diet supplemented with 8 g/kg pomelo peel powder. The following tables and figures are the same as this.

**Table 2 animals-12-03216-t002:** Dietary pomelo peel powder affected growth performance, diarrhea rate, and organ index of piglets.

Items	Treatments			Standard Error of the Mean	*p* Value
	CON	CTC	PPP		
Initial weight (kg)	7.19	7.37	7.84	0.26	0.618
Final weight (kg)	15.05 ^b^	15.14 ^b^	17.82 ^a^	0.57	0.047
Average daily feed intake (g/d)	610.4 ^b^	631.2 ^b^	701.4 ^a^	14.93	0.007
Average daily gain (g/d)	280.6 ^b^	277.7 ^b^	356.0 ^a^	14.04	0.028
F/G	2.21	2.31	2.08	0.08	0.494
Organ indexes (g/kg)					
Liver	26.85	26.89	27.64	0.55	0.827
Thymus	0.45	0.43	0.43	0.01	0.628
Pancreas	1.17	1.25	1.39	0.05	0.205
Spleen	1.53 ^ab^	1.80 ^a^	1.36 ^b^	0.08	0.042
Kidney	4.80	5.65	4.69	0.31	0.439
Diarrhea rate (%)					
Diarrhea rate on the 1st week	26.4	28.6	16.9	0.043	0.51
Diarrhea rate on the 2nd week	28.4	18.5	17.8	0.038	0.482
Diarrhea rate on the 3rd week	36.6 ^a^	8.9 ^b^	10.7 ^b^	0.044	0.017
Diarrhea rate on the 4th week	35.6 ^a^	17.8 ^b^	14.4 ^b^	0.036	0.035

^a, b^ different letter superscripts in the same row were marked on data for significant difference (*p* < 0.05).

**Table 3 animals-12-03216-t003:** Dietary pomelo peel powder affected serum cytokines and immunoglobulin concentrations of weaned piglets.

Items	Treatments			Standard Error of the Mean	*p* Value
	CON	CTC	PPP		
SOD (U/mL)	42.41 ^b^	53.19 ^ab^	68.69 ^a^	4.38	0.039
GSH-Px (U/mL)	82.46 ^b^	72.09 ^b^	105.48 ^a^	5.06	0.011
CAT (U/mL)	7.25 ^b^	5.42 ^b^	11.00 ^a^	0.99	0.032
T-AOC (U/mL)	3.56 ^b^	5.04 ^ab^	6.95 ^a^	0.54	0.044
IL-1β (pg/mL)	215.71 ^a^	189.17 ^a^	131.25 ^b^	10.03	0.000
IL-6 (pg/mL)	238.81	295.5	247.68	18.77	0.482
IL-4 (pg/mL)	16.98 ^b^	22.17 ^a^	23.47 ^a^	1.03	0.015
IL-8 (pg/mL)	23.30 ^a^	13.35 ^b^	7.83 ^b^	2.12	0.003
IL-10 (pg/mL)	35.28	57.58	52.99	4.56	0.132
IL-17 (ng/Ml)	82.61 ^a^	71.00 ^a^	34.10 ^b^	8.55	0.037
IFN-γ (pg/mL)	12.87 ^a^	13.46 ^a^	8.24 ^b^	0.83	0.011
PGE2 (pg/mL)	169.30	231.22	227.46	13.00	0.094
IgA (μg/mL)	18.36^c^	30.60 ^a^	23.67 ^b^	1.35	0.001
IgG (μg/mL)	24.96 ^b^	36.73 ^a^	32.75 ^a^	1.77	0.021
IgM (μg/mL)	70.04	86.86	91.93	4.23	0.800
IGF-I (ng/mL)	51.85 ^b^	91.60 ^a^	88.04 ^a^	5.63	0.002

^a, b, c^ different letter superscripts in the same row were marked on data for significant difference (p < 0.05).

**Table 4 animals-12-03216-t004:** Dietary pomelo peel powder affected pH in gastrointestinal tract and ileum morphology of weaned piglets.

Items	Treatments			Standard Error of the Mean	*p*-Value
	CON	CTC	PPP		
Ph value					
Stomach	4.67 ^a^	3.29 ^b^	3.65 ^b^	0.35	0.006
Duodenum	6.41 ^a^	5.15 ^b^	5.27 ^b^	0.23	0.023
Jejunum	6.22	5.98	5.87	0.10	0.411
Ileum	6.95	6.79	6.59	0.10	0.387
Colon	6.46	6.11	6.18	0.08	0.177
Cecum	6.25	5.92	5.74	0.10	0.107
Ileum morphology					
Villus height (µm)	241 ^b^	296 ^a^	303 ^a^	11.56	0.035
Crypt depth (µm)	210	225	209	7.11	0.635
Villus height/crypt depth	1.15 ^c^	1.32 ^b^	1.45 ^a^	0.04	0.001

^a, b, c^ different letter superscripts in the same row were marked on data for significant difference (*p* < 0.05).

**Table 5 animals-12-03216-t005:** Analysis of microflora diversity of colonic contents in each treatment group.

Items	Treatments			Standard Error of the Mean	*p* Value
	CON	CTC	PPP		
Simpson	0.97	0.99	0.98	0.01	0.342
Chao1	1344.19	1938.11	1656.68	119.03	0.118
ACE	1410.50	1970.71	1719.52	120.00	0.163
Shannon	7.66	8.76	8.36	0.20	0.052

**Table 6 animals-12-03216-t006:** The relative abundance of colonic microflora at phylum level (%).

Items	Treatments			Standard Error of the Mean	*p* Value
	CON	CTC	PPP		
Firmicutes	71.85 ^a^	53.78 ^b^	50.51 ^b^	3.81	0.014
Bacteroidetes	21.14 ^b^	40.77 ^a^	36.03 ^a^	3.44	0.026
Spirochaetes	1.14	0.22	9.42	2.58	0.274
Proteobacteria	1.57	1.94	2.28	0.36	0.769
Tenericutes	0.26	0.08	0.49	0.08	0.101
Cyanobacteria	1.61 ^a^	0.26 ^b^	0.06 ^b^	0.25	0.012
Actinobacteria	0.25	0.33	0.40	0.04	0.233

^a, b^ different letter superscripts in the same row were marked on data for significant difference (*p* < 0.05).

**Table 7 animals-12-03216-t007:** Dietary pomelo peel powder affected the colonic microflora relative abundance at family level (%).

Items	Treatments			Standard Error of the Mean	*p* Value
	CON	CTC	PPP		
*Prevotellaceae*	8.65 ^b^	24.47 ^a^	12.17 ^b^	2.79	0.048
*Veillonellaceae*	13.87 ^a^	17.95 ^a^	5.36 ^b^	2.14	0.013
*Lachnospiraceae*	14.58	15.87	12.81	1.32	0.676
*Ruminococcaceae*	12.15 ^b^	9.81 ^b^	16.50 ^a^	1.11	0.018
*BacteroidalesS24-7*	10.06	6.38	9.60	0.75	0.095
*Clostridiales*	5.20	4.58	5.14	0.51	0.895
*Lactobacillaceae*	7.36 ^a^	1.57 ^b^	5.17 ^a^	0.91	0.018
*Paraprevotellaceae*	2.32	5.91	4.16	0.72	0.250
*Spirochaetaceae*	1.04	0.20	9.27	2.57	0.281

^a, b^ different letter superscripts in the same row were marked on data for significant difference (*p* < 0.05).

**Table 8 animals-12-03216-t008:** Dietary pomelo peel powder affected the prediction function of intestinal flora.

Items	Treatments			Standard Error of the Mean	*p*-Value
	CON	CTC	PPP		
Cellular processes	0.032 ^b^	0.037 ^a^	0.038 ^a^	0.001	0.028
Genetic information processing	0.208	0.206	0.204	0.001	0.286
Environmental information processing	0.134	0.134	0.1401	0.002	0.290
Human diseases	0.007	0.007	0.007	0.001	0.421
Organismal system	0.007	0.007	0.007	0.001	0.165
Metabolism	0.023 ^a^	0.022 ^b^	0.023 ^a^	0.001	0.046

^a, b^ different letter superscripts in the same row were marked on data for significant difference (*p* < 0.05).

**Table 9 animals-12-03216-t009:** Dietary pomelo peel powder affected the metabolism function of colonic flora.

Items	Treatments			Standard Error of the Mean	*p*-Value
	CON	CTC	PPP		
Amino acid metabolism	0.100	0.099	0.097	0.001	0.217
Biosynthesis of other secondary metabolites	0.009	0.010	0.009	<0.001	0.074
Carbohydrate metabolism	0.101 ^b^	0.100 ^b^	0.103 ^a^	0.002	0.022
Energy metabolism	0.058	0.059	0.058	0.001	0.268
Enzyme families	0.022	0.022	0.022	<0.001	0.684
Lipid metabolism	0.027	0.027	0.027	<0.001	0.554
Nucleotide metabolism	0.042	0.042	0.041	0.001	0.343
Glycan biosynthesis and metabolism	0.022	0.022	0.021	0.001	0.560
Terpenoids and polyketides metabolism	0.017	0.016	0.016	<0.001	0.066
Cofactors and vitamins metabolism	0.045	0.046	0.043	0.002	0.135
Other amino acids	0.015	0.015	0.015	<0.001	0.070
Xenobiotics biodegradation and metabolism	0.015 ^a^	0.014 ^b^	0.015 ^a^	0.001	0.007

^a, b^ different letter superscripts in the same row were marked on data for significant difference (*p* < 0.05).

## Data Availability

The 16sRNA sequencing raw data are available at NCBI under the accession number PRJNA814319 by web link (https://www.ncbi.nlm.nih.gov/sra/ PRJNA814319, accessed on 9 March 2022). The rest of the raw data supporting the conclusions of this article will be made available by the authors without any undue reservation.

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
