# Peer review of "Effect of Dietary Pomelo Peel Powder on Growth Performance, Diarrhea, Immune Function, Antioxidant Function, Ileum Morphology, and Colonic Microflora of Weaned Piglets"

_animals, 2022, doi:10.3390/ani12223216_

Round 1
Reviewer 1 Report
This study assess the effect of dietary grapefruit peel powder on growth performance, diarrhea, immune function, antioxidant function, ileum morphology and colonic microflora of weaned piglets. The topic is interesting and novelty. However the manuscript lacks readability. Extensive editing of English language and style required. There are many things must be correct before acceptance of this manuscript.
1. L33. Please give a full name for ‘CAT’. Abbreviations must be defined at their first mention. Please review the full article.
2. Abstract. The abstract should be a total of about 200 words maximum. The abstract should state briefly the purpose of the research, the principal results and major conclusions, not only the purpose and conclusion. Please add the principal results.
3. L 66. Please provide the annual output of grapefruit.
4. L79. Please delated the Prof’s name.
5. L87. ‘7.48 ± 1.19 kg kg’. Please delete one ‘kg’.
6. L86. 2.2. Why use this dose?
7. L116. BW
8. 152. ‘SAS 8.2satatistical’.Please enter a space after 8.2.
9. L153. ‘All data are presented as’ to ‘All data were presented as’
10. Table 2. ‘14.38’ to ‘14.4’.
11. Table 3. The data of T-AOC. Please keep two significant digits after the decimal point
12. L199. ‘in among’, please revise English grammar.
13. L470-471. Please explain the reason GPP and CTC significantly reduce the pH value in duodenum and cecum of weaned piglets
14. L546-549. Please revise this conclusion. Inappropriate expression of ‘replace antibiotics’ .
15. Review the full article. The full text should be written in a consistent format. For example, Line 155, Line 203 and L264. P, P and p.
Author Response
General comments: This study assess the effect of dietary grapefruit peel powder on growth performance, diarrhea, immune function, antioxidant function, ileum morphology and colonic microflora of weaned piglets. The topic is interesting and novelty. However, the manuscript lacks readability. Extensive editing of English language and style required. There are many things must be correct before acceptance of this manuscript. “the results clearly presented” and “the conclusions supported by the results” can be improved.
Authors'responses and locations of the revisions: we sincerely appreciate expert reviewer who have devoted their time to this article and give us many valuable comments and suggestions. Based on the Authors are not from a native-English nation, we entrusted to Editage (www.editage.cn) for English language editing and corrected many mistakes and completely revised the English language. (Please the revised version in heighted) and we had improved “the results clearly presented” and “the conclusions supported by the results”.
Comment 1: L33. Please give a full name for ‘CAT’. Abbreviations must be defined at their first mention. Please review the full article.
Authors'responses and locations of the revisions: we gave a full name (catalase, glutathione peroxidase) for ‘CAT and GSH-Px. Abbreviations had be defined at their first mention in the full article. (Please the abstract in the revised version in heighted)
Comment 2: Abstract. The abstract should be a total of about 200 words maximum. The abstract should state briefly the purpose of the research, the principal results and major conclusions, not only the purpose and conclusion. Please add the principal results.
Authors'responses and locations of the revisions: Thanks. Owing to your suggestion, we shorten the abstract for a total of about 200 words maximum. (Please see abstract in the revised version).
Comment 3: L 66. Please provide the annual output of grapefruit.
Authors'responses and locations of the revisions: Thanks. We had provided the annual output of grapefruit. (Please see lines 66-69 in the revised version).
Comment 4: L79. Please deleted the Prof’s name.
Authors'responses and locations of the revisions: Thanks. we deleted the Prof’s name. (Please see line 66 in the revised version).
Comment 5: L87. ‘7.48 ± 1.19 kg kg’. Please delete one ‘kg’.
Authors'responses and locations of the revisions: Thanks. we deleted it. (Please see line 102 in the revised version).
Comment 6: L86. 2.2. Why use this dose?
Authors'responses and locations of the revisions: Thanks. This dose of pomelo peel powder was based on the previous literature [15].
Comment 7: L116. BW
Authors'responses and locations of the revisions: Thanks. we corrected it as: body weight. (Please see line 163 in the revised version).
Comment 8: 152. ‘SAS 8.2satatistical’. Please enter a space after 8.2.
Authors'responses and locations of the revisions: Thanks. We corrected it. (Please see line 172in the revised version).
Comment 9: L153. ‘All data are presented as’ to ‘All data were presented as’
Authors'responses and locations of the revisions: Thanks. We corrected it. (Please see line 143 in the revised version).
Comment 10: Table 2. ‘14.38’ to ‘14.4’
Authors'responses and locations of the revisions: Thanks. We corrected it. (Please see Table 2 in the revised version).
Comment 11: Table 3. The data of T-AOC. Please keep two significant digits after the decimal point
Authors'responses and locations of the revisions: Thanks. We corrected it. (Please see Table 3 in the revised version).
Comment 12: L199. ‘in among’, please revise English grammar.
Authors'responses and locations of the revisions: Thanks. We corrected it. (Please see the revised version).
Comment 13: L470-471. Please explain the reason GPP and CTC significantly reduce the pH value in duodenum and cecum of weaned piglets
Authors'responses and locations of the revisions: Thanks. Owing to your suggestion, we had explain the reason GPP and CTC significantly reduce the pH value in duodenum and cecum of weaned piglets we modified the relevant sentence. (Please see line L437-442 in the revised version).
Comment 14: L546-549. Please revise this conclusion. Inappropriate expression of ‘replace antibiotics’.
Authors'responses and locations of the revisions: Thanks. Owing to your suggestion, we revised this conclusion. (Please see line L506-515 in the revised version).
Comment 15: Review the full article. The full text should be written in a consistent format. For example, Line 155, Line 203 and L264. P, P and p.
Authors'responses and locations of the revisions: Thanks. We reviewed the full article and corrected all inappropriate format (Please see the revised version).
Reviewer 2 Report
Thank you for an interesting and complex study on the dietary grapefruit peel powder effect on growth performance, diarrhea, immune function, antioxidant function, ileum morphology and colonic microflora of weaned piglets
Although the topic is important for the animal field, certain aspects require clarifications and corrections, as follows:
point 1 line 79 the authors should describe the grapefruit peel powder
point 2 this power underwent any prior in vitro testing and characterization?
point 3 line 128- add the kit description to mention the validity in pigs
point 4 Discussion section - the authors should also refer to other studies on natural products dietary use in pigs (there is a multitude of studies on this topic)
Author Response
General comments: English language and style are fine/minor spell check required. “all the cited references relevant to the research”, “the research design appropriate”, “the methods adequately described” and “the conclusions supported by the results” can be improved. Thank you for an interesting and complex study on the dietary grapefruit peel powder effect on growth performance, diarrhea, immune function, antioxidant function, ileum morphology and colonic microflora of weaned piglets. Although the topic is important for the animal field, certain aspects require clarifications and corrections.
Authors'responses and locations of the revisions: we sincerely appreciate expert reviewer who have devoted their time to this article and give us many valuable comments and suggestions. Based on the Authors are not from a native-English nation, We Entrusted to Editage (www.editage.cn) for English language editing and corrected many mistakes and completely revised the English language. (Please the revised version in heighted) and we had improved “the results clearly presented” and “the conclusions supported by the results”.
Comment 1: line 79 the authors should describe the grapefruit peel powder.
Authors'responses and locations of the revisions: Thanks. We describe the Pomelo peel powder. (Please see lines 64-74 in the revised version).
Comment 2: this power underwent any prior in vitro testing and characterization?
Authors'responses and locations of the revisions: Thanks. The pomelo peel powder was provided by College of Horticulture and Landscape Architecture, Southwest University, Chongqing, China. We didn’t make vitro testing. (Please see lines 99-100 in the revised version).
Comment 3: line 128- add the kit description to mention the validity in pigs
Authors'responses and locations of the revisions: Thanks. We gave more details for kit description. (Please see line 143-147 in the revised version).
Comment 4: Discussion section - the authors should also refer to other studies on natural products dietary use in pigs (there is a multitude of studies on this topic)
Authors'responses and locations of the revisions: Thanks. Owing to your suggestion, we corrected Discussion section. (Please see the revised version).
Reviewer 3 Report
Zeng et al. investigated the effects of dietary supplementation of grapefruit peel powder (GPP) on growth performance, diarrhea, antioxidation and intestinal health of weaned piglets. The experiment is well designed, but the manuscript requires some corrections.
1. line 32, “….and serum antioxidant capacity (P< 0.05) of weaned piglets”, this is more of a conclusion than a description of results, specific antioxidant indices need to be described.
1. line 32-33, “but had little 32 effect on CAT (P > 0.05)…... the CAT and GSH-Px activity of GPP group were significantly higher”. There is no CAT data in this paper. Moreover, if the difference is not significant, multiple comparative analyses cannot be performed.
3. line 38-39, “and improve intestinal microflora of piglets”, this is more of a conclusion than a description of the result. The microbial results need to be more detailed.
4. line 91, why did the authors choose 8 g / kg GPP? What is the rationale for choosing this dose?
5. line 125, Why did the authors measured CAT activity, but not show it in the Table 3?
6.line 125-128, detailed kits information is required.
7. line 130, please define the abbreviation of “H&E”.
8. line 197-198, “…have lower cecal content pH in was significantly than piglets in the CON group (P<0.05).” this sentence is rather difficult to understand.
9. Some important references are recommended to be added, for example:
(1) Tang, X.; Liu, X.; Zhang, K. Effects of Microbial Fermented Feed on Serum Biochemical Profile, Carcass Traits, Meat Amino Acid and Fatty Acid Profile, and Gut Microbiome Composition of Finishing Pigs. Front. Vet. Sci. 2022, 8, 744630. doi: 10.3389/fvets.2021.744630
(2) Tang, X.; Zhang, K.; Xiong, K. Fecal Microbial Changes in Response to Finishing Pigs Directly Fed With Fermented Feed. Front. Vet. Sci. 2022, 9, 894909. doi: 10.3389/fvets.2022.894909
Author Response
Authors'responses and locations of the revisions: we sincerely appreciate expert reviewer who have devoted their time to this article and give us many valuable comments and suggestions. Based on the Authors are not from a native-English nation, We Entrusted to Editage (www.editage.cn) for English language editing and corrected many mistakes and completely revised the English language. (Please the revised version in heighted) and we had improved “the results clearly presented” and “the conclusions supported by the results”.
Comment 1: line 32-33, “but had little 32 effect on CAT (P > 0.05)…... the CAT and GSH-Px activity of GPP group were significantly higher”. There is no CAT data in this paper. Moreover, if the difference is not significant, multiple comparative analyses cannot be performed.
Authors'responses and locations of the revisions: Thanks. We corrected CAT as T-AOC (Please see Abstract in the revised version)
Comment 2: line 38-39, “and improve intestinal microflora of piglets”, this is more of a conclusion than a description of the result. The microbial results need to be more detailed.
Authors'responses and locations of the revisions: Thanks. we gave more detail for The microbial results in abstract. (Please see Abstract in the revised version).
Comment 3: line 91, why did the authors choose 8 g / kg GPP? What is the rationale for choosing this dose?
Authors'responses and locations of the revisions: Thanks. This dose of pomelo peel powder was based on the previous literature [15].
Comment 4: line 125, Why did the authors measured CAT activity, but not show it in the Table 3?
Authors'responses and locations of the revisions: Thanks. this is mistake. We add CAT data (Please see Table 3 in the revised version).
Comment 5: line 125-128, detailed kits information is required.
Authors'responses and locations of the revisions: Thanks. We gave more details for kit description. (Please see lines 143-147 in the revised version)
Comment 6: line 130, please define the abbreviation of “H&E”.
Authors'responses and locations of the revisions: Thanks. We gave full name for “H&E”. (Please see line 148 in the revised version).
Comment 7: line 197-198, “…have lower cecal content pH in was significantly than piglets in the CON group (P<0.05).” this sentence is rather difficult to understand.
Authors'responses and locations of the revisions: Thanks. we revised this sentence. (Please see the revised version).
Comment 8: Some important references are recommended to be added, for example:
(1) Tang, X.; Liu, X.; Zhang, K. Effects of Microbial Fermented Feed on Serum Biochemical Profile, Carcass Traits, Meat Amino Acid and Fatty Acid Profile, and Gut Microbiome Composition of Finishing Pigs. Front. Vet. Sci. 2022, 8, 744630. doi: 10.3389/fvets.2021.744630
(2) Tang, X.; Zhang, K.; Xiong, K. Fecal Microbial Changes in Response to Finishing Pigs Directly Fed With Fermented Feed. Front. Vet. Sci. 2022, 9, 894909. doi: 10.3389/fvets.2022.894909
Authors'responses and locations of the revisions: Thanks. Owing to your suggestion, we cited this two references. (Please see reference 43,44 in the revised version).
Round 2
Reviewer 1 Report
Accept.
Author Response
Thanks
Reviewer 2 Report
The authors submitted a revised form of their manuscript, but the modifications involve the English language and the evaluated product.
All the phrases have an improved English language.
The word "chlorotetracycline" appears 25 times; if the authors refer to chlortetracycline, one of the classical antimicrobial agents, they should correct the word accordingly
The other major modifications of the manuscript involves the evaluated product - "grapefruit" replaced with "pomelo"
These two major modifications do not improve the overall scientifical level of the revised manuscript.
Can the authors explain how both grapefruit and pomelo are able to produce the same results in terms of "growth performance, diarrhea, immune function, antioxidant function, ileum morphology and colonic microflora of weaned piglets"?
Regarding the initial comments sent with the first review:
Comment 1: line 79 the authors should describe the grapefruit peel powder.
Authors'responses and locations of the revisions: Thanks. We describe the Pomelo peel powder. (Please see lines 64-74 in the revised version).
Round 2 - Lines 64-74 do not present a description of pomelo peel powder, and my comment did not refer to a general description, but to a characterization for the tested product.
Comment 2: this power underwent any prior in vitro testing and characterization?
Authors'responses and locations of the revisions: Thanks. The pomelo peel powder was provided by College of Horticulture and Landscape Architecture, Southwest University, Chongqing, China. We didn’t make vitro testing. (Please see lines 99-100 in the revised version).
Round 2 - based on what protocol, in vivo testing is considered prior any in vitro evaluation?
Furthermore, the revised form states - lines 161-162 This dose of pomelo peel powder PPP was based on the previous literature [15]. The 15 reference refers to a murine model, what was the base for the dose calculation?
Comment 3: line 128- add the kit description to mention the validity in pigs
Authors'responses and locations of the revisions: Thanks. We gave more details for kit description. (Please see line 143-147 in the revised version).
Round 2- the authors included the requested information
Comment 4: Discussion section - the authors should also refer to other studies on natural products dietary use in pigs (there is a multitude of studies on this topic)
Round 2 - the modifications involve the English language, the requested modifications are not provided by the authors
Author Response
Section 1: "Responses to the Comments by reviewer 2"
General comments: English language and style are fine/minor spell check required. Are the methods adequately described? Must be improved. Are the results clearly presented? Can be improved. Are all the cited references relevant to the research? Can be improved. Is the research design appropriate? Can be improved. Are the conclusions supported by the results? Can be improved. Does the introduction provide sufficient background and include all relevant references? Can be improved. The authors submitted a revised form of their manuscript, but the modifications involve the English language and the evaluated product. All the phrases have an improved English language.
Authors' responses and locations of the revisions: we sincerely appreciate expert reviewer 2 who have devoted their time to this article and give us many valuable comments and suggestions again. We checked English language and style of the whole manuscript spell and completely revised the English language again. We improved the methods, introduction and results and discussion according to suggestions of Reviewer 2 (Please the revised version marked up using the “Track Changes” function)
Comment 1: The word "chlorotetracycline" appears 25 times; if the authors refer to
chlortetracycline, one of the classical antimicrobial agents, they should correct the word
accordingly
Authors'responses and locations of the revisions: We use” chlortetracycline” in this study. We chlorotetracycline have been corrected as “chlortetracycline”. (Please the revised version marked up using the “TrackChanges” function)
Comment 2: The other major modifications of the manuscript involves the evaluated product -"grapefruit" replaced with "pomelo". These two major modifications do not improve the overall scientifical level of the revised manuscript. Can the authors explain how both grapefruit and pomelo are able to produce the same results in terms of "growth performance, diarrhea,immune function, antioxidant function, ileum morphology and colonic microflora of weaned piglets"?
Authors'responses and locations of the revisions: pomelo mainly refer to the type of “first picture”; grapefruit mainly refer to the type of “second picture”; when we translate the first manuscript we mistaken using “grapefruit”. In fact, we used pomelo peel power in this study, so we corrected term "grapefruit" replaced with "pomelo".
We have corrected the whole manuscript.
Comment 3: Lines 64-74 do not present a description of pomelo peel powder, and my comment did not refer to a general description, but to a characterization for the tested product.
Authors'responses and locations of the revisions: we have provided a characterization of of PPP (Please see L93-97 in the revised version marked up using the “TrackChanges” function)
Comment 4: based on what protocol, in vivo testing is considered prior any in vitro evaluation? Furthermore, the revised form states - lines 161-162 This dose of pomelo peel powder PPP was based on the previous literature [15]. The 15 reference refers to a murine model, what was the base for the dose calculation?
Authors'responses and locations of the revisions: some studied were reported that PPP can be applied to diet in mice [15-16]. We didn’t find literature that reported applied to diet in piglets, so this dose of PPP was based on 1g/kg body weigh/day for mice reported by previous literature [15-16] with slight modifications.
15.Kawabata, A.; Hung, T. V.; Nagata, Y.; Fukuda, N.; Suzuki, T. Citrus kawachiensis peel powder reduces intestinal barrier defects and inflammation in colitic mice. J. Agri. and Food Chem. 2018, 66 (42), 10991-10999. doi:10.1021/acs.jafc.8b03511.
16.Hu, M.; Zhang, L.; Ruan, Z.; Han, P.; Yu, Y. The Regulatory Effects of Citrus Peel Powder on Liver Metabolites and Gut Flora in Mice with Non-Alcoholic Fatty Liver Disease (NAFLD). Foods. 2021, 10(12), 3022. doi: 10.3390/foods10123022.
Comment 5: (round 1) Discussion section - the authors should also refer to other studies on natural products dietary use in pigs (there is a multitude of studies on this topic). (round 2) the modifications involve the English language, the requested modifications are not provided by the authors
Authors'responses and locations of the revisions: Thanks. We have provided other studies on natural products dietary use in pigs in Discussion sections. (Please see the revised version marked up using the “TrackChanges” function).

Reviewer 3 Report
This manuscript can be accepted.
Author Response
Thanks